# Strong band renormalization and emergent ferromagnetism induced by electron-antiferromagnetic-magnon coupling

T. L. Yu[1], M. Xu[1], W. T. Yang[1], Y. H. Song[1], C. H. P. Wen[1], Q. Yao[1], X. Lou[1], T. Zhang[1,2,3], W. Li[1], X. Y. Wei[1], J. K. Bao[4], G. H. Cao[4], P. Dudin[5], J. D. Denlinger[6], V. N. Strocov[7], R. Peng[1,2] ✉, H. C. Xu[1] ✉ & D. L. Feng[1,2,3,8] ✉

The interactions between electrons and antiferromagnetic magnons (AFMMs) are important for a large class of correlated materials. For example, they are the most plausible pairing glues in high-temperature superconductors, such as cuprates and iron-based superconductors. However, unlike electron-phonon interactions (EPIs), clear-cut observations regarding how electron-AFMM interactions (EAIs) affect the band structure are still lacking. Consequently, critical information on the EAIs, such as its strength and doping dependence, remains elusive. Here we directly observe that EAIs induce a kink structure in the band dispersion of $Ba_{1-x}K_xMn_2As_2$, and subsequently unveil several key characteristics of EAIs. We found that the coupling constant of EAIs can be as large as 5.4, and it shows strong doping dependence and temperature dependence, all in stark contrast to the behaviors of EPIs. The colossal renormalization of electron bands by EAIs enhances the density of states at Fermi energy, which is likely driving the emergent ferromagnetic state in $Ba_{1-x}K_xMn_2As_2$ through a Stoner-like mechanism with mixed itinerant-local character. Our results expand the current knowledge of EAIs, which may facilitate the further understanding of many correlated materials where EAIs play a critical role.

Electron–boson interactions belong to the most fundamental microscopic processes in solids, which are responsible for various fascinating properties. For example, electron–phonon interactions (EPIs) could result in conventional superconductivity or charge density waves[1,2], whereas the high-temperature superconductivity in cuprate and iron-based superconductors is proposed to be related to the interactions between electrons and antiferromagnetic (AFM) spin fluctuations, i.e., AFM magnons (AFMM)[3–5]. The electron–boson interactions "dress" the electrons up, and convert the electrons into quasiparticles. Theoretically, the effects of electron–boson interaction can be described by a complex self-energy of the electronic structure[6,7], which is determined by both the electron–boson matrix

[1]Laboratory of Advanced Materials, State Key Laboratory of Surface Physics and Department of Physics, Fudan University, 200438 Shanghai, P. R. China. [2]Shanghai Research Center for Quantum Sciences, 201315 Shanghai, P. R. China. [3]Collaborative Innovation Center of Advanced Microstructures, 210093 Nanjing, China. [4]Department of Physics, Zhejiang University, 310027 Hangzhou, P. R. China. [5]Diamond Light Source, Harwell Science and Innovation Campus, Didcot OX11 0DE, UK. [6]Advanced Light Source, Lawrence Berkeley National Laboratory, 1 Cyclotron Road, Berkeley, CA 94720-8229, USA. [7]Swiss Light Source, Paul Scherrer Institut, CH-5232 Villigen, PSI, Switzerland. [8]Hefei National Laboratory for Physical Science at Microscale, CAS Center for Excellence in Quantum Information and Quantum Physics, and Department of Physics, University of Science and Technology of China, 230026 Hefei, P. R. China. ✉e-mail: pengrui@fudan.edu.cn; xuhaichao@fudan.edu.cn; dlfeng@ustc.edu.cn

element $g$ (i.e., interaction potential) and the material-specific bosonic structure[8]. Experimentally, the band structure is renormalized, and sometimes an abrupt distortion, i.e., a kink, in the otherwise smooth band dispersion can be detected by angle resolved photoemission spectroscopy (ARPES)[6,7]. Based on the ARPES data, one can extract the electron self-energy and the electron–boson coupling constant $\lambda$, which characterizes the total renormalization strength and directly gives $T_c$ in BCS theory.

The kinks induced by electron–phonon interaction have been detected in various materials, while the coupling constant $\lambda$ is found to be less than 1 in most metals[9–11]. In contrast, the contribution of AFMM to the electron self-energy has not been experimentally identified, although electron-AFMM interactions (EAIs) holds the key to the general understanding of many correlated phenomena like unconventional superconductivity. Although kink features have been widely observed in the dispersion of many cuprate superconductors[12–19], the origin has been controversial, as the energy scales of kinks are found in both the magnetic resonances[20] and the phonon modes[9,21,22]. The presence of kinks in the heavily overdoped regime without many AFM fluctuations[9,10,23,24] disfavors a magnon origin. Moreover, the energy scale of AFM spin fluctuations is comparable to that of the electronic dispersion near Fermi energy ($E_F$) in cuprates or iron-based superconductors[25–27], which would lead to a renormalization of the entire band rather than a kink near $E_F$[28,29]. Therefore, one cannot isolate or identify the EAIs in the retrieved total self-energy.

To reveal the characteristics of the EAIs, one requires a compound that has both robust AFMM excitations and relatively large electron bandwidth. This dilemma condition is usually difficult to fulfil, while $Ba_{1-x}K_xMn_2As_2$, an isostructure material for 122-type Fe-based superconductors[30], could be an ideal system. On one hand, $BaMn_2As_2$ is an AFM insulator (AFI) with a Néel temperature $T_N = 625$ K. The magnetic moments of $Mn^{2+}$ ions in $BaMn_2As_2$ point along the $c$ axis, forming a G-type AFM order [Fig. 1b]. K doping turns $Ba_{1-x}K_xMn_2As_2$ into a metal [Fig. 1a], while the $T_N$ and magnetic moments persists[31,32]. Inelastic neutron scattering studies find strong AFMM excitations in heavily doped samples[33]. On the other hand, the electron bandwidth could be large in $Ba_{1-x}K_xMn_2As_2$ as observed in the parent compound $BaMn_2As_2$[34]. At $x > 0.19$, it shows an emergent ferromagnetic (FM) ground state with the FM moments in the $ab$-plane contributed by As-4$p$ orbital rather than the canting of Mn AFM moments[32,35,36], while the underlying mechanism of the emergent FM is unknown. Here, by conducting ARPES studies on $Ba_{1-x}K_xMn_2As_2$, we directly observe a kink feature of clean EAIs origin. We subsequently unveil its doping-dependent and temperature dependent behaviors. Its renormalization on the electronic structure is unusually strong, which drives the emergent ferromagnetic state through a Stoner-like mechanism.

## Results and discussions
### Electronic structure of $Ba_{1-x}K_xMn_2As_2$
Figure 1d–f shows ARPES results measured using vacuum ultra-violet (VUV) photons, which resolve the Fermi surface structure of $Ba_{0.7}K_{0.3}Mn_2As_2$ in its three-dimensional Brillouin zone. In the Γ-X-M plane measured with 78 eV photons [Fig. 1e], the photoemission intensity mapping shows a large pocket $\alpha$ in rounded square shape centered at Γ. The $\alpha$ pocket is absent in the ZRA plane [Fig. 1f] and highly dispersive along $k_z$, forming a drum-shaped Fermi surface [Fig. 1d]. The spectral intensity at zone center shows two-dimensional character and persists along $k_z$ [Fig. 1d–f], which is contributed by bands $\beta$ and $\beta'$ whose band top locates near $E_F$. Except for some variation in the spectral weight due to the photoemission matrix element, the Fermi surfaces measured using VUV photons are consistent with the soft X-ray ARPES results (Section 1 of Supplemental Materials), confirming the bulk nature of the measured bands. By calculating the Fermi surface volume in the three-dimensional Brillouin zone, we estimated the hole doping based on Luttinger theorem. The estimated hole doping agrees well with the K concentration at various doping levels (Section 2 of Supplemental Materials).

Along Γ-X, the hole-like bands $\alpha$ and $\beta$ are resolved [Figs. 1g–i]. The $\beta'$ band is relatively weaker but observable in the second-derivative spectra [Fig. 1h], which is also reproduced in calculations[34]. Both the Fermi surfaces and band dispersions of $Ba_{0.7}K_{0.3}Mn_2As_2$ roughly agree with the pristine $BaMn_2As_2$ after a chemical potential shift[31,34]. The bandwidth is in the energy scale of 1 eV, which is one order of magnitude larger than that in most Fe-based superconductors, such as the isostructural $Ba_{1-x}K_xFe_2As_2$[37]. The large bandwidth of $Ba_{1-x}K_xMn_2As_2$ provide a clean playground for observing the kinks of electron–boson interactions and retrieving the self-energies.

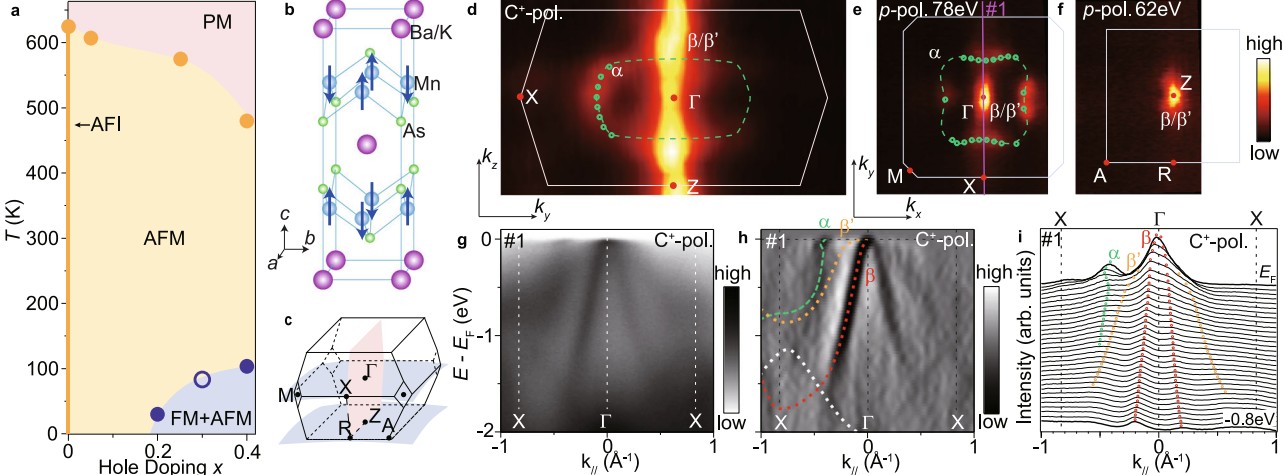

**Fig. 1 | Phase diagram and the basic electronic structure of $Ba_{1-x}K_xMn_2As_2$ (x = 0.3). a** Phase diagram of $Ba_{1-x}K_xMn_2As_2$. The open circle indicates the Curie temperature determined by magnetic susceptibility measurements on our sample (Section 14 of Supplemental Materials). The filled circles indicate data from refs. 32,35,69,70. **b, c** Crystal structure and three-dimensional Brillouin zone of $Ba_{1-x}K_xMn_2As_2$. The blue arrows indicate the alignment of the Mn magnetic moments in the G-type antiferromagnetic (AFM) state. **d–f** Photoemission intensity maps in the high symmetric planes of the Brillouin zone, integrated over an energy window of $E_F \pm 20$ meV. The $k_y - k_z$ map in the ΓXZ plane is measured with $C^+$-polarized (pol.) photons from 58 to 100 eV. **g, h** Photoemission intensity along cut #1 in panel **e** taken by $C^+$-pol. photons and its second derivative with respect to momentum, respectively. The dashed curves indicate the band dispersions. **i** Momentum distribution curves (MDCs) along cut #1. The ARPES data were measured at 30 K, BL5-2 of SSRL.

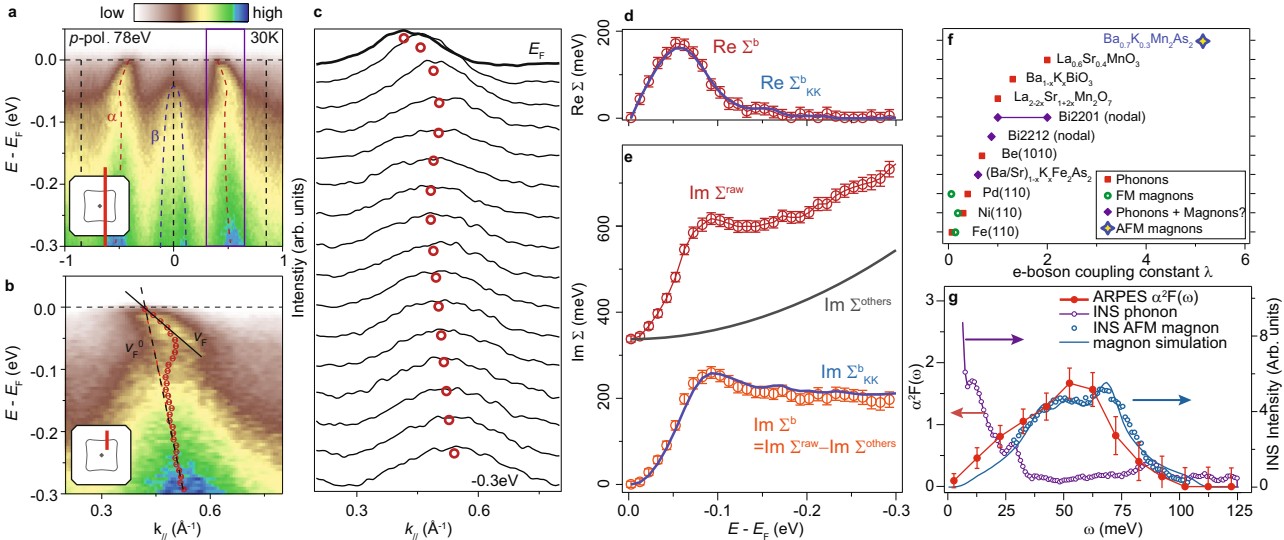

**Fig. 2 | Kink and electron–magnon interactions in $Ba_{0.7}K_{0.3}Mn_2As_2$.**
**a** Photoemission spectra slightly off the zone center, where the $\beta'$ band is outside of the energy window. **b** Zoom-in of the purple rectangular region in panel **a**, overlaid with the MDC peak positions (red circles), fitting of the Fermi velocity $v_F$ (black solid line) and the velocity of large-scale dispersion $v_F^0$ (black dashed line), and the parabolic estimation of the "bare band" dispersion (red dashed curve). A momentum-independent background is subtracted. **c** MDCs of spectra in panel **b**, where the red circles indicate the peak positions from fittings. **d** Re $\Sigma^b$ (red circles) and the KK transformation of Im $\Sigma^b$ in panel **e** (Re $\Sigma^b_{KK}$, blue curve). **e** Im $\Sigma^{raw}$ from the FWHM of MDCs peaks (red circles), KK transformation of Re $\Sigma^b$ (Im $\Sigma^b_{KK}$, blue curve), and the background (Im $\Sigma^{others}$, gray curve). The Im $\Sigma^b$ (orange circles) is

the background-subtracted Im $\Sigma^{raw}$. **f** Reported electron–boson coupling constant obtained from MDC analysis of kinks in Be(1010)[38], Bi2212[39], Bi2201[40], $Ba_{1-x}K_xBiO_3$[11], $La_{0.6}Sr_{0.4}MnO_3$[41], $La_{2-2x}Sr_{1+2x}Mn_2O_7$[42], Fe(110)[43,44], Ni(110)[45], Pd(110)[46], and (Ba/Sr)$_{1-x}K_xFe_2As_2$[47], together with that of $Ba_{0.7}K_{0.3}Mn_2As_2$. **g** Eliashberg function (red circles) estimated from the Im $\Sigma^b$, compared with the phonon (purple circles) and magnon (blue circles) intensity from inelastic neutron scattering (INS) and simulated magnetic intensity (blue curve) on $Ba_{1-x}K_xMn_2As_2$ ($x = 0.25$)[33]. The error bars in panel **d**, **e**, and **g** are propagated from the standard deviation of fittings and the instrumental momentum resolution (see Section 6 of Supplemental Materials for more details). The ARPES data were measured at 30 K, BL4.0.3 of ALS.

## Strong electron–magnon interactions in $Ba_{1-x}K_xMn_2As_2$

The $\alpha$ band shows a kink around the binding energy $E_B = 50$ meV, as shown by the photoemission image and MDCs [Fig. 2a–c, and Section 4 of Supplementary Materials]. The bands $\beta$ and $\beta'$ also show similar features (Section 5 of Supplementary Materials). The kink in electronic dispersion is a general signature of electron–boson interactions[7]. At the kink feature of band $\alpha$, the Fermi velocity $v_F$ is renormalized from the velocity of large-scale dispersion $v_F^0$ by a factor of ~6.1 [Fig. 2b]. The corresponding electron–boson coupling constant reaches $\lambda = v_F^0/v_F - 1 \simeq 5.1$. The coupling constant of the kink in $Ba_{0.7}K_{0.3}Mn_2As_2$ is colossal and larger than those of most of the reported kinks [Fig. 2f][11,38–47], except those near the anti-nodal region of cuprates that are enhanced by superconductivity[18,19].

To retrieve the self-energy of band $\alpha$, we conducted numerical fittings on the MDCs of Fig. 2a using Lorentzian peaks and a constant background (see Section 6 of Supplementary Materials for details). The momentum location of Fig. 2a is set slightly off the zone center to avoid the influence from $\beta'$ band to the fittings. The real part of the self-energy contributed by the kink-related bosonic modes is obtained by Re $\Sigma^b(E, \mathbf{k}) = E(\mathbf{k}) - E_0(\mathbf{k})$, where $E_0(\mathbf{k})$ is a parabolic estimation of the "bare band" dispersion. The imaginary part of the self-energy is obtained by $|\text{Im } \Sigma^{raw}(E)| = |v_F^0| \times \text{FWHM}/2$, where FWHM is the full-width at half-maximum from fitting the MDCs. Corresponding to the kink energy of $Ba_{0.7}K_{0.3}Mn_2As_2$, the Re $\Sigma^b$ peaks at -50 meV [Fig. 2d], where the imaginary part of the self-energy Im $\Sigma^{raw}$ also shows a step [Fig. 2e], indicating the energy scale of the kink-related bosonic modes. After subtracting Im $\Sigma^{raw}$ by a background (Im $\Sigma^{others}$) that accounts for extrinsic broadening and other effects of electron correlations (Section 6 of Supplementary Materials,[48,49]), we obtain the Im $\Sigma^b$ that is contributed by the kink-related bosonic modes. The Im $\Sigma^b$ matches well with the Kramers-Kronig (KK) transformation of Re $\Sigma^b$ [Im $\Sigma^b_{KK}$ in Fig. 2d]. Vice versa, the KK transformation of Im $\Sigma^b$ [Re $\Sigma^b_{KK}$ in Fig. 2c] also matches well with Re $\Sigma^b$. The KK conjugation between the Re $\Sigma^b$

from dispersion and Im $\Sigma^b$ from FWHM helps to confine the fitting processes and supports the self-consistency of the extracted self-energies.

The conventional Landau quasiparticle picture requires Im $\Sigma \ll E_B$ near $E_F$, while in $Ba_{0.7}K_{0.3}Mn_2As_2$ the Im $\Sigma^b$ rises quickly below $E_F$ and exceeds $E_B$ [Fig. 2e]. The large magnitude of self-energy comes from the strength of the electron–boson interactions and indicates a deviation from conventional quasiparticle picture. Besides, although the simulated ARPES spectra based on the quasiparticle spectral function reproduces the kink feature (Section 7 of Supplementary Materials), it does not give the high-energy spectral weight at $E_B >$ 100 meV. The high-energy spectral weight increases at higher $E_B$ [Fig. 2a, b], which is opposite to the expectation for the high-energy increasing of Im $\Sigma^{raw}$ in a conventional quasiparticle picture. The deviations from conventional quasiparticle picture also exist in the ARPES spectra of cuprates[50] and manganites[51]. Similar to the method for cuprates with large coupling constant[18], the self-energy analysis here is a semi-conventional quasiparticle-like approach. The high-energy spectral weight that deviates from conventional quasiparticle picture could relate with a polaronic metallic state induced by strong correlations with bosons like phonons or magnetic excitations[51,52].

The obtained Im $\Sigma^b$ and Re $\Sigma^b$ consistently indicate a bosonic energy scale that extends up to -100 meV and peaks at ~50 meV [Fig. 2d, e]. Based on the extracted self-energy, we can roughly estimate the Eliashberg function $\alpha^2F(\omega)$ of the kink by $\alpha^2F(\omega) = \frac{\partial \text{Im}\Sigma(E)}{\pi\partial E}|_{E=\omega}$ (see Section 6 of Supplementary Materials for details), which characterizes the corresponding bosonic density of states (DOS) weighted by their effective interactions with electrons[7]. As shown in Fig. 2f, the energy scale of $\alpha^2F(\omega)$ is distinct from the phonon energies below 25 meV according to the inelastic neutron scattering (INS) on $Ba_{1-x}K_xMn_2As_2$ ($x = 0.25$)[33], and the phonon spectra calculated for $Ba_{1-x}K_xMn_2As_2$ ($x = 0$ and 0.5) (Section 11 of Supplementary Materials,[53–56]). EPI could exist below 25 meV, while it is not relevant to

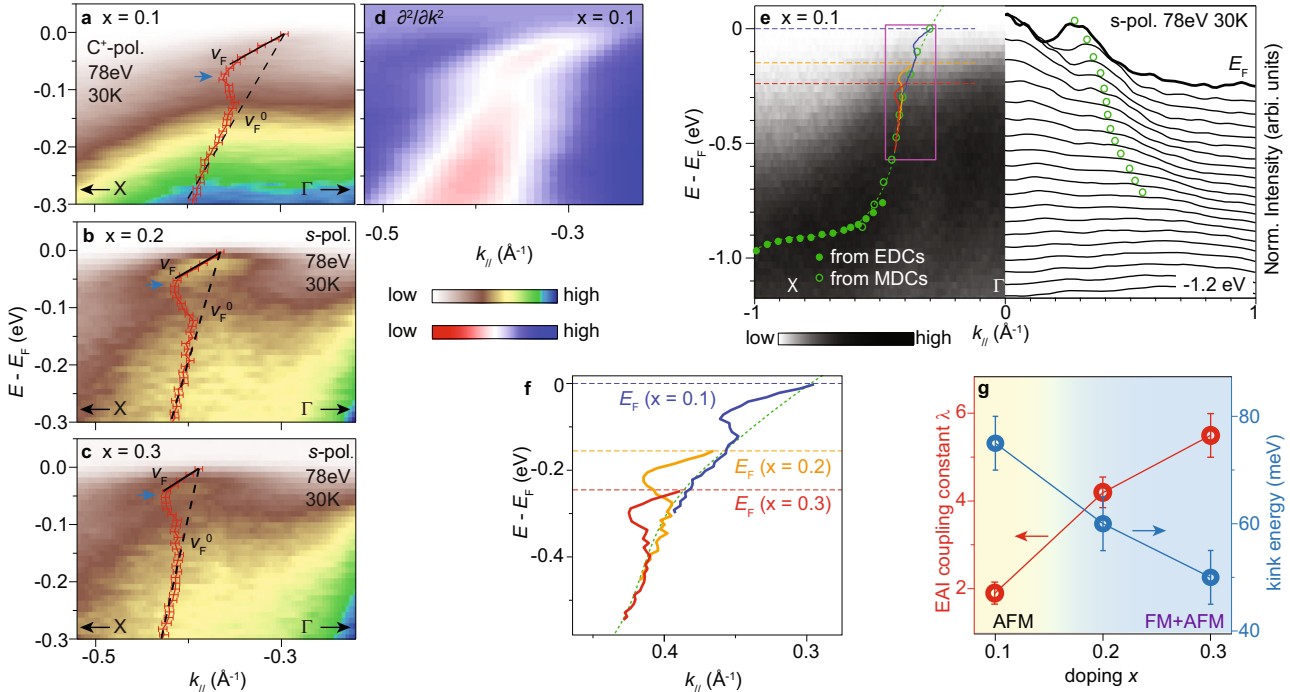

**Fig. 3 | Doping dependence of electron-magnon interactions.**
**a–c** Photoemission intensity near the Fermi crossing of band $\alpha$ for dopings $x = 0.1$, 0.2, and 0.3, respectively. The overlaid dispersion is obtained by fitting the MDCs. The solid lines and dashed lines illustrate the $v_F$ and $v_F^0$, respectively. A momentum-independent background is subtracted. The error bars are based on the standard deviation of fittings and the instrumental momentum resolution. **d** Corresponding second derivative of panel **a**. **e** Photoemission intensity and MDCs along $\Gamma X$ for

doping $x = 0.1$, overlaid with the dispersion of band $\alpha$ (green dashed curve), and the dispersions from panels **a–c** shifted in energy following the rigid band picture. **f** Dispersions in the purple rectangle of panel **e**. **g** Coupling constant and kink energies as a function of dopings. The color background illustrates the transition between the AFM phase and FM+AFM coexisting phase. The error bars are based on the standard deviation of fittings and the instrumental energy resolution. The ARPES data were measured at 30 K, BL5-2 of SSRL.

the kink at 50 meV. On the other hand, the energy scale of $\alpha^2F(\omega)$ matches that of magnon spectra of $Ba_{0.75}K_{0.25}Mn_2As_2$ [Fig. 2f], which has been resolved by the INS magnetic intensity and reproduced by Heisenberg-model-based simulations (Section 12 of Supplemental Materials,[33,57]). After considering the momentum restriction of single-magnon absorption/emission processes, the simulated $\alpha^2F(\omega)$ also exhibits the same energy scale (Section 13 of Supplemental Materials,[33]). The consistent energy scale provides compelling evidence that the kink feature in $Ba_{1-x}K_xMn_2As_2$ is due to the strong interactions between itinerant electrons and AFMMs.

Kinks are present at band $\alpha$ for dopings $x = 0.1$, 0.2, and 0.3 of $Ba_{1-x}K_xMn_2As_2$ [Fig. 3a–d]. Since there is no ferromagnetic transition at doping levels lower than $x = 0.19$, it further supports that the kink originates from AFM magnons rather than FM magnons. The large-scale band dispersion of $\alpha$ is hole-like near $E_F$ with respect to the zone center and becomes relatively steep at $-E_B = 0.3–0.7$ eV [Fig. 3e]. When the doping increases, the dispersion of band $\alpha$ shifts with chemical potential in a rigid band manner (Section 9 of Supplemental Materials). Though the steep dispersion below the kink of $x = 0.3$ [Fig. 3c] is reminiscent of the "waterfall" feature in cuprates[58–60], its doping-dependent energy scale in $Ba_{1-x}K_xMn_2As_2$ is distinct from the fixed energy scale of "waterfall" features at various dopings[59–61]. As the chemical potential lowers across the hole-like dispersion [Fig. 3f], the "bare band" velocity $v_F^0$ below the kink increases from $2.68 \pm 0.11$ eV·Å at $x = 0.1$ to $7.41 \pm 0.43$ eV·Å at $x = 0.3$. The coupling constant is obtained for each doping based on $\lambda = v_F^0/v_F - 1$, which increases with doping by approximately three times from $x = 0.1–0.3$ [Fig. 3g]. The Fermi velocity $v_F$ is nearly doping independent [Fig. 3a–d], which is likely a coincidence due to that both the $v_F^0$ and $\lambda$ increase with doping. The energy of kink feature gradually shifts towards lower energy with increasing doping [Fig. 3a–c and g], which is consistent with the

doping-dependent energy shift of magnons as revealed by both the INS magnetic intensity and the simulated magnon DOS (Section 13 of Supplemental Materials,[33,57]).

## The emergent ferromagnetic ground state
As the K doping increases, a ferromagnetic state emerges in $Ba_{1-x}K_xMn_2As_2$ at dopings $x > 0.19$ [Fig. 1a and Refs. 32,35], which is proposed to be itinerant from the spin polarization of As-$4p$ holes[35,36]. When $Ba_{0.7}K_{0.3}Mn_2As_2$ enters ferromagnetic phase with decreasing temperature, the quasiparticle-like weight of band $\alpha$ increases and becomes sharper [Fig. 4a]. The enhancing of quasiparticle-like weight across the Curie temperature is reported in many ferromagnetic systems of transition metal compounds[62–64]. On top of a temperature-independent incoherent weight that extends to near $E_F$ [inset of Fig. 4e], the quasiparticle-like weight increases continuously with decreasing temperature [Fig. 4e and g]. The major enhancement takes place at the ferromagnetic transition.

The kink feature persists above the Curie temperature in the spectra of $Ba_{0.7}K_{0.3}Mn_2As_2$ [Fig. 4a, b]. From 30 K to 120 K, the Fermi velocity $v_F$ increases from ~0.55 eV·Å to ~0.83 eV·Å, while the $v_F^0$ is almost fixed around 3.4 eV·Å, indicating larger coupling constant $\lambda$ at lower temperatures. Based on $\lambda = v_F^0/v_F - 1$, the EAIs coupling constant $\lambda$ increases by ~70% from 120 to 30 K [Fig. 4f], while a prominent increase occurs between 100 and 65 K across the ferromagnetic transition. As summarized in Fig. 4f, g, both the quasiparticle-like weight and the EAI coupling constant $\lambda$ follow the emergence of ferromagnetic moment with decreasing temperature. Meanwhile, the emergence of ferromagnetic state with doping is also accompanied by an enhancement of EAI coupling constant [Fig. 3g]. These results indicate an intimate relationship between the ferromagnetic state and the EAIs.

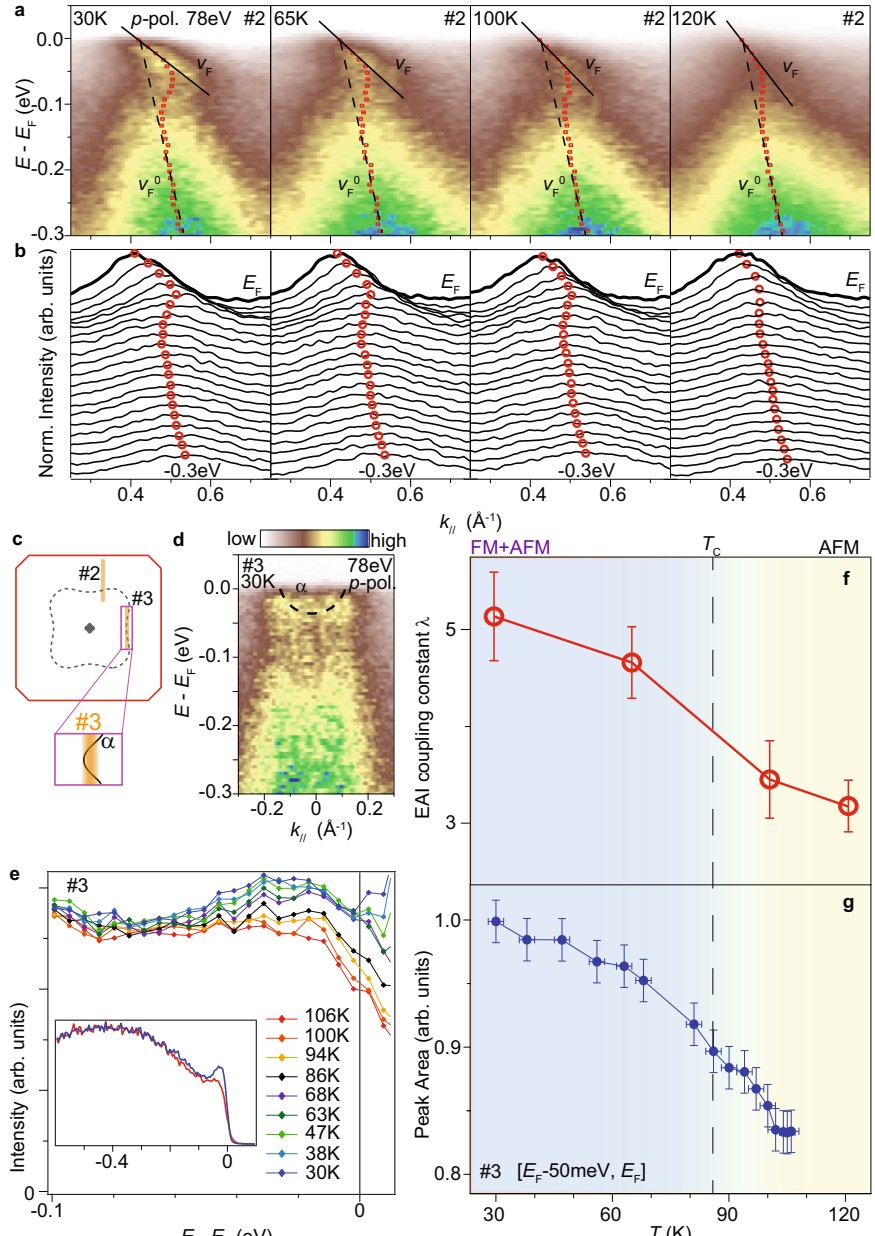

**Fig. 4 | Temperature dependence of Electron–magnon interactions.**
**a** Photoemission intensity along cut #2 of $Ba_{0.7}K_{0.3}Mn_2As_2$ at different temperatures measured at BL4.0.3 of ALS. The solid lines and dashed lines illustrate the the $v_F$ and $v_F^0$, respectively. A momentum-independent background is subtracted.
**b** Normalized MDCs of spectra in panels **a**. The red circles in panels **a** and **b** indicate the band dispersion from fitting the MDCs. **c** Sketch of the Fermi surfaces and photoemission cuts. **d** Photoemission intensity along cut #3 measured at I05 of Diamond, and the dispersion of band $\alpha$ (dashed curve). **e** Integrated spectra of different temperatures along cut #3. The spectra are divided by a resolution-convolved Fermi-Dirac function to remove the thermal effect. The inset shows integrated spectra in a larger energy range. **f, g** Coupling constant and integrated intensity over $[E_F$-50 meV, $E_F]$ as a function of temperature, respectively. The error bars are based on the standard deviation of fittings and the instrumental resolution.

Across the Curie temperature, our ARPES data resolve no detectable exchange splitting in $Ba_{0.7}K_{0.3}Mn_2As_2$ [Fig. 4a]. Even if there is a finite exchange splitting obscured by extrinsic momentum broadening, the estimated upper limit of itinerant magnetic moment is less than 40% of that from magnetic susceptibility measurements (see Section 16 of Supplemental Materials for details). Furthermore, our DFT+U calculation, without considering the EAIs, gives diminishing exchange splitting at As-4$p$ bands either (Section 15 of Supplemental Materials). As doping increases from $x = 0.1$ to $x = 0.3$, the Fermi surface volume evolves as a normal metal rather than a half-metal (Section 2 of Supplemental Materials). These results indicates that the ferromagnetism in $Ba_{1-x}K_xMn_2As_2$ is in mixed itinerant and localized character.

The itinerant holes are likely playing critical roles in the ferromagnetic transition, considering the coincident enhancement of EAIs and ferromagnetic transition with either doping or cooling. The EAIs in $Ba_{1-x}K_xMn_2As_2$ could influence the ferromagnetic transition through enhancing the DOS at $E_F$, noted as $N(E_F)$. Here we estimate the $N(E_F)$ directly by its definition $N(E_F) = \frac{\pi}{4} \int \frac{dS}{4\pi^3} \frac{1}{v_F}$, where the integration goes over the Fermi surfaces (Section 10 of Supplemental Materials). Note that this estimation only involves the size of Fermi surfaces and the Fermi velocities without considering the experimental photoemission spectral weight. Based on the $v_F^0$ without the renormalization by EAIs, the calculated "bare" DOS $N^0(E_F)$ is -0.3 spin$^{-1}$eV$^{-1}$As$^{-1}$. This is similar to previous DFT results that does not meet the Stoner criteria[65]. However,

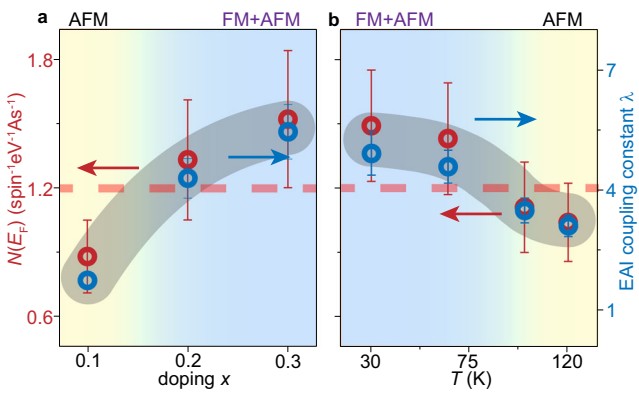

**Fig. 5 | Electron DOS and coupling constant in the temperature-doping phase diagram. a** Electron DOS ($N(E_F)$, red circles) and EAIs coupling constant ($\lambda$, blue circles) as a function of the doping at 30 K. **b** Same as **a** but as a function of the temperature for $x = 0.3$. The color background illustrates the transition between the AFM phase and FM+AFM coexisting phase. The horizontal dashed line illustrates the Stoner criteria assuming $I_{As} \sim 0.8$ eV, to fit the phase boundary of the ferromagnetic phase. The error bars of $\lambda$ are propagated from fitting the Fermi velocities, while those of $N(E_F)$ are based on fitting the Fermi velocities and estimating the Fermi surface volumes (Sections 2 and 10 of Supplemental Materials).

after considering the renormalization by EAIs, the $N(E_F)$ estimated from renormalized $v_F$ is greatly enhanced with respect to $N^0(E_F)$. The $N(E_F)$ further increases with doping or cooling [Fig. 5], following the doping dependence and temperature dependence of the coupling constant $\lambda$. It becomes most enhanced in the ferromagnetic regime, reaching $1.52 \pm 0.32$ spin$^{-1}$eV$^{-1}$As$^{-1}$ at $x = 0.3$, which is nearly three times of that estimated by DFT calculations at an even higher doping $x = 0.4$[65]. As summarized in Fig. 5, whenever $N(E_F)$ is increased roughly to ~1.2 spin$^{-1}$eV$^{-1}$As$^{-1}$ either by doping or cooling, the broad ferromagnetic transition takes place. These results suggest that a Stoner-like mechanism is at work for driving the ferromagnetic ground state through As-$4p$ holes. The As-$4p$ states could obtain a localized character through the Mn-As hybridization that projects the the As-$4p$ orbital to the flat bands at high binding energies[34], while the itinerant As-$4p$ holes near $E_F$ are strongly renormalized by the EAIs to reach the DOS of Stoner criteria and give rise to the ferromagnetism in Ba$_{1-x}$K$_x$Mn$_2$As$_2$ with mixed itinerant and localized character.

The EAI is one of the fundamental interactions in condensed matter. Our ARPES study on Ba$_{1-x}$K$_x$Mn$_2$As$_2$ provides a clean observation of EAI-induced kink. Our findings illustrate quantitatively how the EAIs can renormalize the electron band and help to induce ferromagnetism in Ba$_{1-x}$K$_x$Mn$_2$As$_2$, which reveals a unique pathway to realizing emergent ground states like ferromagnetism by the strong interaction between electrons and AFM order. It also offers an unprecedented opportunity to directly study the self-energy and Eliasgberg function $\alpha^2F(\omega)$ of EAIs, and our quantitative analysis reveals various important facts about the EAIs.

- The coupling constant $\lambda$ of EAIs in Ba$_{1-x}$K$_x$Mn$_2$As$_2$ ($x = 0.3$) can reach as high as 5.4, which is much higher than most other known electron–boson interactions in metals. The large coupling constant demonstrates that the effect of the EAIs can be overwhelmingly strong and could induce emergent ground states. The high spin $S = 5/2$ of Mn and relatively small bandwidth of magnons give rise to high magnon DOS, which could be responsible for the large coupling constant. Nevertheless, the underlying behavior of EAI should be universal and could be applied to other systems like cuprates and iron-based superconductors.
- The EAIs vary strongly with both temperature and doping, even if the AFM order is relatively robust. The EPIs usually decrease at

higher carrier concentrations due to screening. In contrast, the EAIs in Ba$_{1-x}$K$_x$Mn$_2$As$_2$ is enhanced by three times from $x = 0.1$–$0.3$ with increasing hole concentration. Based on simulations (Section 13 of Supplemental Materials), the increased scattering phase space at larger Fermi surfaces could explain the enhanced coupling constant at higher hole dopings.

These features of EAIs put constrains on the theories. The extracted electron self-energy and Eliashberg function $\alpha^2F(\omega)$ are critical for revealing the electron–magnon matrix element $g$ and other microscopic characteristics of EAIs at a more quantitative level, which encourages future inelastic scattering studies on the magnon spectra with absolute value. As these characteristics of the EAIs could be general in other materials with AFM excitations, such as high-temperature superconductivity in cuprates and iron-based compounds, our results would facilitate further understanding of the related emergent phenomena.

## Methods

The Ba$_{1-x}$K$_x$Mn$_2$As$_2$ single crystals used in this study were grown by the flux method as described elsewhere[32,66]. The chemical composition and K doping level were determined by electron probe microanalysis (EPMA). The magnetic susceptibility was measured by Quantum Design Dynalcool system. VUV-ARPES experiments were performed at beamlines BL5-2 of the Stanford Synchrotron Radiation Light Source (SSRL), I05 of the Diamond Light Source, and BL4.0.3 of the Advanced Light Source (ALS). The energy and angular resolutions were set at 15 meV and 0.1°, respectively. Soft X-Ray ARPES experiments were performed at ADRESS of the Swiss Light Source[67]. The energy and angular resolutions were set at 80 meV and 0.1°, respectively. All samples were cleaved in situ and measured under a vacuum better than $8 \times 10^{-11}$ mbar. The homogeneity of the cleaved surfaces was confirmed by ARPES, STM, STS, and EPMA (Sections 1–3 of Supplemental Materials and refs. 34,68,69). Each group of data for doping dependence or temperature dependence study was taken in the same beamline facility. Furthermore, the data taken at different beamlines shows consistent electronic structure and spectral features. For doping $x = 0.3$ and temperature $T = 30$ K, the data taken at SSRL, Diamond, and ALS give identical values of $k_F$ and $v_F$ within the experimental resolution. Further analysis gives the similar EAI coupling constant $\lambda$ and density-of-staes $N(E_F)$.

## Data availability

The relevant data supporting our key findings are available within the article and the Supplementary Information file. All raw data generated during our current study are available from the corresponding authors upon reasonable request.

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

## Acknowledgements

We gratefully acknowledge the valuable discussion with Prof. J. Zhao, and experimental support of Dr. D. H. Lu, Dr. M. Hashimoto, Dr. Y. B. Huang, Dr. Z. Sun, Dr Z. T. Liu, and Dr. D. W. Shen. We thank the Diamond Light Source for time on beamline I05, the Advanced Light Source (U.S. DOE contract no. DE-AC02-05CH11231) for access to beamline 4.0.3, the Stanford Synchrotron Radiation Light Source for the access to beamline 5-2, and the Swiss Light Source for time on SX-ARPES endstation of beamline ADRESS. Some preliminary data were taken at National Synchrotron Radiation Laboratory (NSRL, China) and BLO3U at Shanghai Synchrotron Radiation Facility. This work is supported in part by the National Natural Science Foundation of China (Grants No. 11888101, 12074074, and 11790312), the National Key R&D Program of the MOST of China (Grants No. 2017YFA0303004 and 2016YFA0300200), Project supported by Shanghai Municipal Science and Technology Major Project (Grant No. 2019SHZDZX01), and Shanghai Rising-Star Program (20QA1401400).

## Author contributions

T.Y., M.X., C.W., Q.Y., X.L., R.P., and H.X. collected the ARPES data. T.Y., H.X., R.P., and D.F. analyzed the ARPES data. W.Y. and T.Z. measured the STM. Y.S. measured the EPMA. W.L. and X.W. conducted the phonon and electronic band calculations. H.X. performed the simulations. J.B. and G.C. provided the single-crystal samples. P.D., J.D., and V.S. assisted the experiments at beamlines. T.Y., H.X., R.P., and D.F. wrote the paper. H.X., R.P., and D.F. are responsible for the infrastructure, project direction, and planning. All authors have discussed the results and the interpretation.

## Competing interests

The authors declare no competing interests.

## Additional information

**Correspondence and requests** for materials should be addressed to R. Peng, H. C. Xu or D. L. Feng.

