## [Peer Review File · Nature Communications]

Reviewers' Comments:

Reviewer #1:

Yu and colleagues investigate the strong and abrupt renormalization of the band in (Ba,K)Fe₂As₂ including the temperature and doping dependence. With supporting information from other experimental results and theoretical considerations, they associate the observed abrupt renormalization, so-called kink, as a result of the electron-antiferromagnetic-magnon interaction (EAI). They also successfully analyzed the renormalization in detail, the estimation of the coupling strength through extracting the self-energy which exhibits beautiful causality between real and imaginary parts. Further, the consequence of the renormalization, the enhancement of the density of state at the Fermi level, is discussed as the source of the ferromagnetic moment in the K doped case.

The quantitative estimation of the density of state enhancement upon doping is mostly convincing, also the related discussion. In fact, the estimation is free from the origin of the band renormalization as it only accounts for the Fermi surface volume and Fermi velocity which can be extracted from the observed band structure itself.

On the other hand, the claim that one might be doubtful about is the EAI origin of the band renormalization. Other than EAI, the possible origin would be a band gap opening due to the band folding or coupling with other bosonic modes including phonon. The latter possibility could be simply denied. As the authors considered, the energy scale of phonons simply does not match the energy scale of the observed kink. Also, as there is no known ordering other than antiferromagnetism, expecting any other order-associated collective excitation other than magnon is extremely unlikely. The former possibility is also hard to be the case. One could imagine the presence of possible short-range order that generates band folding although there are no experimental clues on it yet. However, in such a case, the band crossing point between the original and folded band should be fixed in terms of momentum position so does the renormalization, which is not the case. In that regard, the summary of band renormalization given in Fig. 3(f) is critical as it demonstrates that the band renormalization for all doping levels consistently occurs in reference to the bare band of the green dashed line, once the rigid band shift is applied.

Therefore, assigning the observed kink as a consequence of EAI is definitely convincing. Actually, a beautiful match between the extracted real part of self-energy and that retrieved from the imaginary part of self-energy (vice versa) strongly indicates the interaction origin of the kink and the qualitative consistency between the ARPES Eliashberg function and the inelastic neutron scattering results supports the EAI as a corresponding interaction.

I believe the present work will be regarded as the seminal work that directly reveals the EAI experimentally. The aspects that could be informative for future theoretical effort are also provided including the large coupling constant. All the other details are carefully considered and given in a well-organized manner. Considering all, I recommend the publication of the present manuscript, strongly and delightfully.

Reviewer #2:

Remarks to the Author:

This paper reports on the observation of low-energy dispersion kink in a material system in a robust antiferromagnetic (AFM) state. This kink was found to be associated with a strong renormalization of the electronic band dispersion of a magnitude that is unprecedented among all the dispersion kinks observed so far in condensed matter systems (more than twice stronger than the previous records). This renormalization was clearly shown to be consistent with the result of electron-AFM magnon interaction rather than that of electron-phonon interaction in the system. A systematic study of the renormalization in terms of the doping dependence and temperature dependence suggested its intimate connection with the ferromagnetic (FM) state that sets in at low temperatures. It was proposed that an increasing AFM-induced renormalization leads to an increase in the density of states at the Fermi level that drives the system into a FM state. The entire experimental case looks quite clean and convincing to me, which can potentially become another textbook example of the electron-boson coupling in solids, not to mention its unique position as the first example of an AFM-induced kink. I recommend the publication of this paper after the following minor issues are properly addressed.

1. Please provide references to support the statement made in Lines 46-47: "... which would lead to a renormalization of the entire band rather than a kink near the Fermi energy (EF)". A clear account for this point would facilitate the understanding of why a dispersion kink due to the electron-AFM magnon coupling had been elusive previously as opposed to its striking manifestation reported here.

2. A strong mode coupling tends to break up a single band into two branches, which appears to be the case of this study. An alternative to this interpretation is that the two branches could well be due to two separate band dispersions. Although the observed temperature dependence of the dispersion kink seems to have ruled this out, the case would be strengthened further if the authors could show that no separate band formation can be observed associated with the low-energy branch regardless of the experimental conditions (such as photon energies other than 78 eV).

3. Is this the first experimental case for a AFM-driven transition into a FM state? In any case, the AFM-induced kink formation provides a unique pathway to realizing such a transition. This perspective could be interesting enough to mention.

Reviewer #3:

Remarks to the Author:

Yu and coauthors have conducted a complete set of experimental and theoretical studies on electron-AFMM interactions (EAIs) in $\text{Ba}_{1-x}\text{K}_x\text{Mn}_2\text{As}_2$. Notably, this study provides a clean playground with direct observation of EAIs-induced kink in the band dispersion. This awarded the authors an exceptional opportunity to investigate if EAIs could induce ground states in correlated materials.

With the kink structure's temperature and doping dependence, the authors argued that EAIs in $\text{Ba}_{1-x}\text{K}_x\text{Mn}_2\text{As}_2$ could be responsible for the emergent ferromagnetic ground state through colossal band renormalization and stoner-like mechanism.

Overall, this manuscript is a pleasant read with a clear and well-structured discussion of both experimental results and theoretical calculations. EAIs is one of the most prevailing proposals responsible for high-temperature superconductivity in cuprate and iron-based superconductors. The fact that EAIs could induce a ferromagnetic ground state demonstrated in this study with strong evidence is an important milestone for the condensed matter physics community to deepen our understanding of correlated materials.

Considering its high-quality investigations and scientific soundness, I would recommend this manuscript's publication with a few minor revisions as follows,

1. Line 80: "Along Γ -X, the bands α , β , and β' are all resolved" [Figures 1(i)].

If we examine the ARPES MDCs waterfall plot in Figure 1(i) closely, it is not convincing that band β' is resolved. Indeed, in the supplementary material section 5, the authors themselves also ignored band β' in the fittings.

2. In figure 2c caption: "MDCs of panel (b). The red circles track the local maxima."

Are the red circles in panels (c) and (b) trace the same MDC peak positions obtained from the data fittings? Regardless, the statement "The red circles track the local maxima" in panel (c) is not true from what is plotted.

The same issue is also with figures 4 a and b.

3. In figure 2e caption: background ($\text{Im } \Sigma$ others, green curve).

The background curve is plotted as the color grey, instead of green, in panel (e).

4. In figure 2e caption, a brief description of the error bar in panel 2e is missing.

5. In figure 2g, the legend for the blue solid line (Heisenberg-model-based simulation) is missing.

6. In figure 3a,b,c, and g caption, a brief description of the fitting error bar is missing

7. In figure 4, there are two sets of data: #2 and #3. The temperature data points are much finer for cut #3, whereas cut #2 only has 4 temperatures. Since #2 is what we are more interested in, especially around transition Curie temperature, more temperature data points are preferred. Are there any experimental challenges for cut #2? With only 4 data points in figure 4 for cut #2, the

authors' statement on line 175 "with a sharper increase across the ferromagnetic transition" is not supported by the data.

8. Figure 5, how is the error bar for each data point calculated? It is common to leave the details in the supplementary material, but it is necessary to give a brief description in the main text for general audiences.

9. In the experimental methods section, line 242 to line 244 mention that ARPES data are collected from three different beamlines and beamtimes. The authors must provide explicit descriptions for all data shown in this manuscript of their sources, especially, the doping and temperature dependence data (attach beamline info to the figure caption for all figures). How the systematics of the experiments was achieved with data from three beamtimes and facilities? The authors must give enough justifications.

Reply to reviewers:

Following are the one-to-one responses to the reviewers' comments. The reviewers' comments are in blue color, while our responses are in black.

Reviewer #1:

Yu and colleagues investigate the strong and abrupt renormalization of the band in (Ba,K)Fe₂As₂ including the temperature and doping dependence. With supporting information from other experimental results and theoretical considerations, they associate the observed abrupt renormalization, so-called kink, as a result of the electron-antiferromagnetic-magnon interaction (EAI). They also successfully analyzed the renormalization in detail, the estimation of the coupling strength through extracting the self-energy which exhibits beautiful causality between real and imaginary parts. Further, the consequence of the renormalization, the enhancement of the density of state at the Fermi level, is discussed as the source of the ferromagnetic moment in the K doped case.

The quantitative estimation of the density of state enhancement upon doping is mostly convincing, also the related discussion. In fact, the estimation is free from the origin of the band renormalization as it only accounts for the Fermi surface volume and Fermi velocity which can be extracted from the observed band structure itself.

Reply: We thank the reviewer for summarizing our results, and pointing out that our quantitative estimation of the density of states and the related discussion are convincing.

On the other hand, the claim that one might be doubtful about is the EAI origin of the band renormalization. Other than EAI, the possible origin would be a band gap opening due to the band folding or coupling with other bosonic modes including phonon. The latter possibility could be simply denied. As the authors considered, the energy scale of phonons simply does not match the energy scale of the observed kink. Also, as there is no known ordering other than antiferromagnetism, expecting any other order-associated collective excitation other than magnon is extremely unlikely. The former possibility is also hard to be the case. One could imagine the presence of possible short-range order that generates band folding although there are no experimental clues on it yet. However, in such a case, the band crossing point between the original and folded band should be fixed in terms of momentum position so does the renormalization, which is not the case. In that regard, the summary of band renormalization given in Fig. 3(f) is critical as it demonstrates that the band renormalization for all doping levels consistently occurs in reference to the bare band of the green dashed line, once the rigid band shift is applied.

Therefore, assigning the observed kink as a consequence of EAI is definitely convincing. Actually, a beautiful match between the extracted real part of self-energy and that retrieved from the imaginary part of self-energy (vice versa) strongly indicates the interaction origin of the kink and the qualitative consistency between the ARPES Eliashberg function and the inelastic neutron scattering results supports the EAI as a corresponding interaction.

Reply: We thank the reviewer for the careful reading and professional comments, especially excluding possible scenario other than EAs.

I believe the present work will be regarded as the seminal work that directly reveals the EA experimentally. The aspects that could be informative for future theoretical effort are also provided including the large coupling constant. All the other details are carefully considered and given in a well-organized manner. Considering all, I recommend the publication of the present manuscript, strongly and delightfully.

Reply: We thank the reviewer for the strong recommendation.

Reviewer #2 (Remarks to the Author):

This paper reports on the observation of low-energy dispersion kink in a material system in a robust antiferromagnetic (AFM) state. This kink was found to be associated with a strong renormalization of the electronic band dispersion of a magnitude that is unprecedented among all the dispersion kinks observed so far in condensed matter systems (more than twice stronger than the previous records). This renormalization was clearly shown to be consistent with the result of electron-AFM magnon interaction rather than that of electron-phonon interaction in the system. A systematic study of the renormalization in terms of the doping dependence and temperature dependence suggested its intimate connection with the ferromagnetic (FM) state that sets in at low temperatures. It was proposed that an increasing AFM-induced renormalization leads to an increase in the density of states at the Fermi level that drives the system into a FM state. The entire experimental case looks quite clean and convincing to me, which can potentially become another textbook example of the electron-boson coupling in solids, not to mention its unique position as the first example of an AFM-induced kink. I recommend the publication of this paper after the following minor issues are properly addressed.

Reply: We thank the reviewer for pointing out the significance of our findings and the strong recommendation.

1. Please provide references to support the statement made in Lines 46-47: "... which would lead to a renormalization of the entire band rather than a kink near the Fermi energy (EF)". A clear account for this point would facilitate the understanding of why a dispersion kink due to the electron-AFM magnon coupling had been elusive previously as apposed to its striking manifestation reported here.

Reply: We thank the reviewer for pointing out the missing of references. In the revised manuscript, we have added two references [PSSB. 200404959 (2004) and Nat. Commun. 15769 (2017)] to this statement. The added references show that, when the energies of bosonic modes are higher than the electronic energy scale, the entire electronic band are renormalized in a polaronic manner rather than forming a kink. The renormalization factor of the electronic bandwidth depends on the electron-boson coupling strength.

2. A strong mode coupling tends to break up a single band into two branches, which appears to be the case of this study. An alternative to this interpretation is that the two branches could well be due to two separate band dispersions. Although the observed temperature dependence of the dispersion kink seems to have ruled this out, the case would be strengthened further if the authors could show that no separate band formation can be observed associated with the low-energy branch regardless of the experimental conditions (such as photon energies other than 78 eV).

Reply: We thank the reviewer for the helpful comments. Following the reviewer's suggestion, here in Fig.R1 we show the data taken using different photon energies other than 78 eV. The data show no separate band formation, while the kink feature is always observable. This excludes the alternative interpretation of two branches of bands. We have added the above figure and discussion as Section 4 of Supplemental Materials.

Fig.R1 (a) Photoemission intensity map in the k_y - k_z plane, where the red curves illustrate the momentum cuts corresponding to 72eV and 76eV photon energies. (b)-(d) Photoemission spectra taken with photon energies 76eV, 74eV, and 72eV, respectively. (e)-(g) MDCs around the kink feature of band α , corresponding to the region illustrated by red rectangles in panels b-d. The red circles trace the dispersion of band α . The data were taken at 30K.

3. Is this the first experimental case for a AFM-driven transition into a FM state? In any case, the AFM-

induced kink formation provides a unique pathway to realizing such a transition. This perspective could be interesting enough to mention.

Reply: We thank the reviewer for the helpful comments. Yes, as far as we know, this is the first experimental case for an e-AFMM-coupling-driven transition into a FM state. Following the reviewer's suggestion, we have emphasized that "...reveals a unique pathway to realizing emergent ground states like FM by the strong interaction between electrons to AFM order" in the concluding part of the revised manuscript.

We thank the reviewer for all the valuable comments and suggestions, which have helped to improve our manuscript and made it more rigorous.

Reviewer #3 (Remarks to the Author):

Yu and coauthors have conducted a complete set of experimental and theoretical studies on electron-AFMM interactions (EAls) in $\text{Ba}_{1-x}\text{K}_x\text{Mn}_2\text{As}_2$. Notably, this study provides a clean playground with direct observation of EAls-induced kink in the band dispersion. This awarded the authors an exceptional opportunity to investigate if EAls could induce ground states in correlated materials.

With the kink structure's temperature and doping dependence, the authors argued that EAls in $\text{Ba}_{1-x}\text{K}_x\text{Mn}_2\text{As}_2$ could be responsible for the emergent ferromagnetic ground state through colossal band renormalization and stoner-like mechanism.

Overall, this manuscript is a pleasant read with a clear and well-structured discussion of both experimental results and theoretical calculations. EAls is one of the most prevailing proposals responsible for high-temperature superconductivity in cuprate and iron-based superconductors. The fact that EAls could induce a ferromagnetic ground state demonstrated in this study with strong evidence is an important milestone for the condensed matter physics community to deepen our understanding of correlated materials.

Reply: We thank the reviewer for acknowledging the importance of our work and regarding our finding that EAls could induce an FM ground state as "an important milestone".

Considering its high-quality investigations and scientific soundness, I would recommend this manuscript's publication with a few minor revisions as follows,

1. Line 80: "Along Γ -X, the bands α , β , and β' are all resolved" [Figures 1(i)]. If we examine the ARPES MDCs waterfall plot in Figure 1(i) closely, it is not convincing that band β' is resolved. Indeed, in the supplementary material section 5, the authors themselves also ignored band β' in the fittings.

Reply: We agree with the reviewer that the band β' is in weaker intensity and is not clear in the MDC plot of Fig.1i. However, it is observable in the second derivative with respect to energy, as shown in Fig.R2a. Moreover, the band β' is also observed in the ARPES study of the parent compound BaMn_2As_2

(Zhang et. al., Phys. Rev. B 94, 155155 (2016)) and is present in band structure calculations (Zhang et. al., Phys. Rev. B 94, 155155 (2016) and section 14 of the supplementary information).

To clarify, in our data analysis including Supplemental Material Section 5, the band β' is not ignored, but it is outside of the energy window because the momentum cut is slightly off the zone center, as illustrated in Fig.R2b. We choose this momentum cut so that the MDC fittings could be simpler without involving β' .

To avoid confusion, we cite the calculations when describing β' , and added the explanation of the chosen momentum cut in the caption.

Fig.R2 (a) Second derivative with respect to energy of the ARPES spectra in Fig.1g. (b) Sketch of the momentum cuts of Fig.1g (cut1) and Fig.2a (cut2), and the isoenergy contours of bands β and β' .

2. In figure 2c caption: "MDCs of panel (b). The red circles track the local maxima."

Are the red circles in panels (c) and (b) trace the same MDC peak positions obtained from the data fittings? Regardless, the statement "The red circles track the local maxima" in panel (c) is not true from what is plotted.

The same issue is also with figures 4 a and b.

Reply: Yes, the circles trace the same MDC peak positions obtained from the data fittings. We thank the reviewer for pointing out this issue. We have corrected the corresponding captions in the revised manuscript.

3. In figure 2e caption: background (Im Σ others, green curve). The background curve is plotted as the color grey, instead of green, in panel (e).

4. In figure 2e caption, a brief description of the error bar in panel 2e is missing.

5. In figure 2g, the legend for the blue solid line (Heisenberg-model-based simulation) is missing.

6. In figure 3a,b,c, and g caption, a brief description of the fitting error bar is missing

Reply: We thank the reviewer for the careful reading and helpful comments of points 3-6. We have fixed these mistakes and added the descriptions on error bars.

7. In figure 4, there are two sets of data: #2 and #3. The temperature data points are much finer for cut #3, whereas cut #2 only has 4 temperatures. Since #2 is what we are more interested in, especially around transition Curie temperature, more temperature data points are preferred. Are there any experimental challenges for cut #2? With only 4 data points in figure 4 for cut #2, the authors' statement on line 175 "with a sharper increase across the ferromagnetic transition" is not supported by the data.

Reply: The temperature dependent experiments on cut #3 and #2 are aimed at spectral weight and self-energy analysis, respectively. It is easier to get enough statistics for the spectral weight analysis. Especially, the cut #3 is at a grazing angle along the Fermi surface, which improves the statistical counts of the quasiparticle weight and allows more data points of temperatures. On the other hand, it requires much higher data quality with better statistics to fit the MDCs for self-energy analysis. The cut #2 is nearly perpendicular to the Fermi surface for cleaner dispersion, but the relatively weak intensity of quasiparticle weight further increases the data collection time. For the reliability of our analysis, we only take high statistical scans at four temperatures along cut #2. This is the experimental challenge we face in the current stage.

We agree with the reviewer that it is not rigorous to claim a "sharper increase" from four data points. In the revised manuscript, we have modified the description to "..., while a prominent increase occurs between 100K and 65K across the ferromagnetic transition."

8. Figure 5, how is the error bar for each data point calculated? It is common to leave the details in the supplementary material, but it is necessary to give a brief description in the main text for general audiences.

Reply: We thank the reviewer for the helpful suggestions. The error bars of the coupling constant λ are propagated from those of fitting the Fermi velocities. The error bars of the density of states $N(E_F)$ are from combining those of Fermi velocities and those of estimating the Fermi surface volumes (Section 2 and 10 of Supplementary Materials). We have added these descriptions in the revised manuscript.

9. In the experimental methods section, line 242 to line 244 mention that ARPES data are collected from three different beamlines and beamtimes. The authors must provide explicit descriptions for all data shown in this manuscript of their sources, especially, the doping and temperature dependence data (attach beamline info to the figure caption for all figures). How the systematics of the experiments was achieved with data from three beamtimes and facilities? The authors must give enough justifications.

Reply: We agree with the reviewer that the facilities where the data are taken should be specified and their systematics should be justified. The data in fig. 1 and fig. 3 were obtained in SSRL, the data in fig.

2 and fig. 4a-b (cut #2) were obtained in ALS, while the data in fig. 4d-e (cut#3) were obtained in Diamond. We have included the above information into our revised captions.

Each group of data for doping dependence or temperature dependence study was taken in the same beamline facility. Furthermore, the data of BKMA taken at different beamlines shows consistent electronic structure and spectral features. For doping $x=0.3$ and temperature $T=30\text{K}$, the data taken at SSRL, Diamond and ALS give identical values of Fermi momentum k_f and Fermi velocity v_f within the experimental resolution. Further analysis gives the similar EAI coupling constant λ and density-of-states $N(E_f)$. The above points guarantee the systematics of the study involving multiple beamlines. We have added these points into the methods section.

Finally, we thank the reviewer for the careful reviewing and helpful comments, which help us to improve our manuscript and make it more rigorous.

Reviewers' Comments:

Reviewer #2:

Remarks to the Author:

I am pleased to see all my comments addressed properly and would like recommend the publication of this paper.

Reviewer #3:

Remarks to the Author:

Thank you for the response. The updated manuscript looks great to me. I would recommend its publication in its current form.

Reply to Reviewers:

Reviewer #2 (Remarks to the Author):

I am pleased to see all my comments addressed properly and would like recommend the publication of this paper.

Reply: We thank the reviewer's affirmative to our last reply and the recommendation of the publication.

Reviewer #3 (Remarks to the Author):

Thank you for the response. The updated manuscript looks great to me. I would recommend its publication in its current form.

Reply: We thank the reviewer's positive comments to our last response and to the updated manuscript, and the recommendation of the publication.